# OpenReview forum: "DRAGON: Guard LLM Unlearning in Context via Negative Detection and Reasoning"
_ICLR.cc/2026/Conference — ICLR 2026 Poster_

### Official Review · Reviewer_aVh4 · 2025-10-24

**Soundness:** 3
**Presentation:** 3
**Contribution:** 3
**Rating:** 4
**Confidence:** 4

**Summary:**

This paper introduces DRAGON (Detect–Reasoning Augmented GeneratiON), a framework for black-box unlearning in large language models (LLMs).
Unlike prior training-based unlearning methods (e.g., TOFU, RMU, NPO-RT, FLAT) that require access to both forget and retain data and involve fine-tuning, DRAGON proposes a training-free, in-context alternative that guards deployed LLMs before inference.

**Strengths:**

* Addresses a real deployment pain point: unlearning without retraining for black-box LLMs.
* Effective and scalable: once trained, the guard model generalizes across models and tasks.
* Comprehensive evaluation: nine LLMs, multiple domains, and both synthetic and realistic datasets.
* Good ablations: CoT necessity, detection variants, continual unlearning, robustness under attack.

**Weaknesses:**

* Over-reliance on refusal behavior. - DRAGON’s “forgetting” may correspond to censorship, not true knowledge deletion. The paper occasionally conflates the two.
* Detection dependency. - The unlearn store and paraphrase-based matching may not scale or generalize to high-entropy data (e.g., private identifiers unseen during training).
* Latency and cost. - Despite claims of low overhead, Table 18 shows inference latency > 600 ms for guarded prompts—non-trivial for interactive systems.

**Questions:**

* How is “forgetting” conceptually distinct from “refusal to answer”? Could DRAGON actually preserve memorized data internally while just refusing to reveal it?
* How sensitive is performance to the choice of CoT guard model (e.g., 7 B vs 13 B)?
* Are DDS/DUS correlated with human judgments of stability, or are they purely mathematical heuristics?

---

> ### Author Response · Authors · 2025-11-25
> **Response (1/2) Weakness 1, 2, 3 and Question 1**
>
> Thank you for your valuable and detailed evaluation of our study and we would like to provide some clarification to address your concerns.
>
> **W1 & Q1 Over-reliance on refusal behavior.**
>
> In our formulation, “forgetting” refers to preventing the model from using or revealing the forgotten information, not necessarily erasing it from the parameters. This is consistent with recent work on training-free unlearning [1, 2, 3], which is suitable for settings where weight editing is impossible.
>
> In API-based or frozen-model deployments, now the dominant mode for commercial LLMs, the base model acts as a closed, integrated system. In such settings: model weights cannot be modified, and retraining is infeasible, but organizations still require the ability to forbid certain knowledge from being generated (e.g., private records, harmful content, proprietary material). Thus, DRAGON’s detector-guard pipeline represents a practical mechanism of knowledge “forgetting” at the system level, even though the underlying parameters remain unchanged.
>
> **DRAGON is not only refusing to answer.**  To avoid sole refusal behavior, especially for private-data unlearning, our guard does not default to “I don’t know.” Instead, the underlying model may generate fabricated but contextually plausible alternative content directed by the in-context intervention, such as fictionalized entities or safe proxy information, ensuring that the responses remain informative, coherent, and consistent with the user’s query while avoiding the disclosure of real private data. This leads to constructive and consistent safety behavior rather than blunt censorship. This is one of the key roles of our guard model, and the consistency score reported in Table 4 provides quantitative evidence that the model does not simply refuse; instead, it generates coherent, contextually appropriate alternative responses.
>
> Our method explicitly detects and intercepts prompts related to forgotten content, and modifies model behavior to avoid generating responses tied to that information. In doing so, we fulfill the core objective of unlearning: preventing the model from retaining or revealing specific knowledge [4].
>
> [1] In-Context Unlearning: Language Models as Few-Shot Unlearners
>
> [2] Large Language Model Unlearning via Embedding-Corrupted Prompts
>
> [3] GUARD: Generation-time LLM Unlearning via Adaptive Restriction and Detection
>
> [4] Large Language Model Unlearning
>
> **W2 Detection Dependency.**
>
> Our detection module does not rely solely on the unlearn store or paraphrase-based matching. Instead, it has a dual-layer architecture that is explicitly designed to handle unseen private information. It integrates: A learned scoring model that captures high-level prompt features to assess alignment with unlearning objectives. A similarity-based metric that computes prompt-to-store embedding distances for second verification. These two signals are combined into a unified confidence score, enabling robust and adaptive thresholding to handle distributional shifts and paraphrased attacks. This dual-layer structure forms a lightweight yet effective algorithm for scalable and accurate detection.
>
> Table 5 demonstrates the effectiveness and generalization of our detection module, which achieves nearly perfect detection performance on TOFU dataset. The trained scoring model can generalize well to unseen data. Appendix D.6 further evaluates the detector under diverse adversarial conditions, including paraphrasing, adversarial prompts, language mix attacks, jailbreaks, and typo perturbations, showing strong robustness across all settings.
>
> **W3: Latency and Cost.**
>
> We acknowledge that our approach introduces a modest increase in inference-time cost compared to standard generation. However, the >600 ms are computationally inexpensive. For non-forget-related prompts, the detection module runs once, and no further intervention is triggered. Thus, the inference latency remains effectively the same as standard LLM inference for the vast majority of input queries. For forget-related prompts, safety becomes the top priority. In such cases, a modest latency increase is acceptable, particularly for sensitive or regulated domains where safety overweighs speed.
>
> Our method strikes a practical balance between efficiency and safety, enabling training-free unlearning with minimal deployment overhead, especially in real-world settings where model internals are inaccessible (e.g., commercial APIs) or continual unlearning is required. We believe **the modest additional cost introduced by the detector and guard model is justified by the effectiveness, safety and reliability they provide compared with other unlearning methods.**

---

> ### Author Response · Authors · 2025-11-25
> **Response (2/2) Question 2, 3**
>
> **Q2 Sensitivity study on the choice of CoT guard model.**
>
> Our method is **robust to the choice of the CoT guard model**, even when the models differ substantially in architecture, scale, and reasoning capabilities. Because the Llama-3 family does not provide a 13B variant, we evaluate with Llama-2-13B-hf, which is not a reasoning-optimized model. We further include Qwen2.5-14B-Instruct, which is a similar-scale alternative. Finally, we add the o3 model as a zero-shot baseline that does not need to be finetuned.
>
> Across TOFU-1% and TOFU-10% (Table 1 and Table 2), we observe:
>
> - DS is consistently strong across guard models except Llama-2-13b-hf.
> - MU remains almost identical regardless of guard model choice.
> - KFR and KRR stay high and stable across both smaller and larger guard models, except for the zero-shot o3.
>
> These results demonstrate that performance **does not depend on a specific guard model size or family**. Even when using a weaker or non-CoT-optimized guard model (Llama-2-13B-hf), DRAGON preserves both unlearning quality and utility compared with other strong baselines, but not as good as using more latest models. In summary, DRAGON maintains stable performance across diverse guard models: 8B vs. 14B, Llama vs. Qwen, and even zero-shot o3, showing that our method is robust and not sensitive to the precise guard model selection. An 8B-level model is already sufficient to generate useful CoT instructions.
>
> The o3 model is used only as a simple reference baseline to understand the potential performance; in practice, it may not be suitable for private-data unlearning scenarios because calling an external API could risk leaking sensitive information.
>
> **Table 1: Sensitivity analysis of DRAGON under different guard models for TOFU-1%.**
>
> | Guard Model | DS(↓) | MU | KFR | KRR |
> | --- | --- | --- | --- | --- |
> | Llama3.1-8B-instruct | 21.40 | 0.6337 | 0.98 | 0.88 |
> | Llama-2-13b-hf | 33.08 | 0.6340 | 0.98 | 0.87 |
> | Qwen2.5-14B-Instruct | 23.68 | 0.6340 | 0.98 | 0.88 |
> | Zero-shot o3 | 27.04 | 0.6340 | 0.88 | 0.89 |
>
> **Table 2: Sensitivity analysis of DRAGON under different guard models for TOFU-10%.**
>
> | Guard Model | DS(↓) | MU | KFR | KRR |
> | --- | --- | --- | --- | --- |
> | Llama3.1-8B-instruct | 26.50 | 0.6337 | 1.00 | 0.90 |
> | Llama-2-13b-hf | 31.50 | 0.6333 | 0.98 | 0.88 |
> | Qwen2.5-14B-Instruct | 26.24 | 0.6336 | 0.99 | 0.85 |
> | Zero-shot o3 | 29.20 | 0.6337 | 0.99 | 0.85 |
>
> **Q3: Are DDS/DUS correlated with human judgments of stability, or are they purely mathematical heuristics?**
>
> DDS and DUS are not arbitrary heuristics; they are designed to capture aspects of stability that correspond to human-interpretable behavior in continual unlearning.
>
> **How DUS is calculated.**
>
> DUS is derived from the model utility score across sequential unlearning rounds. A method that preserves utility well over multiple requests, such as DRAGON, whose post-unlearning utility remains nearly identical to the original model, naturally achieves a high DUS. This aligns with human expectations: a model that continues to answer normal questions well (quantified as high model utility) after many unlearning steps is perceived as more stable.
>
> **How DDS is calculated.**
>
> DDS measures the stability of unlearning trade off across multiple requests. If a method consistently removes the targeted knowledge while maintaining utility at each step, it achieves a higher DDS. This directly corresponds to how humans judge stability: methods that reliably forget what they should forget, while retaining general capability (quantified as low deviation score), are considered more dependable.
>
> **Correlation with human judgment.**
>
> On TOFU (1%, 5%, 10%), both forget-utility trade-off (DS) and utility preservation (MU) are directly interpretable indicators of model quality. DDS and DUS operationalize these intuitive human assessments into quantitative metrics. The stability of generation across the continual unlearning process is reflected precisely in how DS and MU evolve over the three unlearning rounds. Since the model unlearned on TOFU-10% represents the final state after the continual unlearning process, we analyze its generated outputs in detail. Our qualitative inspection shows that the TOFU-10% unlearned model behaves as intended: it responds normally and accurately to non-forget-related queries, while producing refusals or safe, fabricated alternatives for forget-related prompts. These behaviors are quantitatively reflected in the DDS and DUS.
>
> Overall, DDS and DUS are mathematically defined but intentionally aligned with human-interpretable notions of robustness and stability in continual unlearning.

---

### Official Review · Reviewer_ZPsu · 2025-10-29

**Soundness:** 3
**Presentation:** 4
**Contribution:** 3
**Rating:** 6
**Confidence:** 4

**Summary:**

This paper introduces DRAGON, a framework for unlearning in LLMs that operates without fine-tuning the base model or requiring access to retain data. It focuses on practical scenarios like privacy protection, removal of harmful knowledge, and compliance with regulations such as GDPR. DRAGON uses a lightweight detection module to identify "forget-worthy" prompts based on paraphrased negative data stored in an unlearn store, combining similarity metrics with a trained scoring model for harmful content. Upon detection, it routes the prompt through a fine-tuned CoT guard model to generate reasoning instructions, which are prepended to the input for in-context intervention, enabling refusals or redirections. The framework supports both sample unlearning and concept unlearning. Experiments on datasets like WMDP and TOFU show DRAGON achieving near-random guessing accuracy on forget sets with high RQ scores, while preserving MMLU performance, outperforming baselines across models from Zephyr-7B to Mixtral-47B.

**Strengths:**

- The experimental evaluation is thorough and well-designed, with results across nine diverse LLMs on three tasks, including ablation details in Appendix C and comparisons to strong baselines like RMU and ICUL+ using both standard metrics and the novel RQ/DDS/DUS, demonstrating rigorous validation of scalability and generalizability.
- DRAGON's design supports continual unlearning without cumulative utility degradation, as evidenced by low DDS and high DUS in sequential TOFU unlearning steps, making it suitable for dynamic environments where unlearning requests arrive over time.
- It scales efficiently to black-box LLMs without parameter modifications, incurring no additional fine-tuning costs for larger models, and performs well on copyrighted content unlearning, showing versatility across privacy, harm, and IP tasks.

**Weaknesses:**

- Fine-tuning the CoT guard model on a fixed dataset of 1,000 question-CoT pairs limits generalization to out-of-distribution unlearning tasks.
- The threshold-based triggering is adaptive but requires manual tuning of hyperparameters, which could introduce instability in real-world deployments without automated calibration, and the paper lacks sensitivity analysis on these values across different LLM sizes.

**Questions:**

- The paper mentions using GPT-4o for CoT dataset curation in harmful knowledge tasks; is there a risk of privacy leakage when relying on external APIs for sensitive unlearning?
- How does DRAGON's in-context intervention compare to ALKN's dynamic masking on task vectors for mitigating accumulative errors in continual unlearning [1]? In what ways does DRAGON's training-free base model design outperform or underperform GRU's gradient projection method in balancing unlearning and retention [2]?

[1] Wuerkaixi, Abudukelimu, et al. "Adaptive localization of knowledge negation for continual llm unlearning." Forty-second International Conference on Machine Learning. 2025.

[2] Wang, Yue, et al. "Gru: Mitigating the trade-off between unlearning and retention for large language models." arXiv e-prints (2025): arXiv-2503.

---

> ### Author Response · Authors · 2025-11-25
> **Response (1/2) Weakness 1, 2**
>
> We sincerely appreciate the reviewer’s time and effort in reading our paper and offering thoughtful suggestions and constructive feedback.
>
> **W1: Generalization of CoT Guard Model to out-of-distribution unlearning tasks.**
>
> The goal of fine-tuning the CoT guard model is not to memorize dataset-specific patterns, but to teach the model how to produce appropriate CoT safety instructions when given a private-information query. In other words, the fine-tuning step helps the guard model learn the general reasoning template, rather than learning any dataset-specific content. This training objective naturally supports generalization to new, out-of-distribution private-record unlearning tasks, as the model applies the learned pattern to any newly detected private-related question. Note that for each unlearning category, such as private information and harmful knowledge, we have separate guard models.
>
> **Empirical evidence of generalization on the TOFU Dataset.** For the private record unlearning category, we evaluate generalization across three subtasks: TOFU-1%, TOFU-5%, and TOFU-10%.
>
> In Table 4 from the paper, comparing DRAGON (ours) with DRAGON w/o CoT, we observe substantial improvements in both Deviation Score and Consistency Score across all subtasks.
>
> These results confirm that the guard model trained on the pairs generalizes well across TOFU subtasks with varying forget proportions.
>
> **Additional evaluation on an OOD private information dataset.** To further address the reviewer’s concern, we additionally tested the guard model on an OOD personal-information QA dataset, BLUR [1],  that is not included in TOFU and differs significantly in style and content.
>
> Assuming these samples are correctly detected as forget-related, we apply our private-record guard model to generate CoT instructions. To assess generalization and CoT quality, we design a rubric-based evaluation and use GPT-5 as the evaluator. Our rubric assesses two dimensions: 1) policy correctness: whether the reasoning correctly identifies the query as a private-information request and aligns with safety policy; 2) logical coherence: whether the CoT explains why refusal is needed in a logically consistent and faithful manner.
>
> **Table 1 Evaluation of the guard model’s generalization across in-distribution and out-of-distribution tasks. The high-quality CoT Rate is calculated based on the LLM judge.**
>
> |  | TOFU | BLUE |
> | --- | --- | --- |
> | High-quality CoT Rate | 0.95 | 0.88 |
>
> Results show that our guard model maintains strong generalization: the proportion of high-quality CoT refusals remains high on BLUR despite its distributional shift. This provides further evidence that the detection-and-guard framework generalizes across diverse private-information tasks easily.
>
> [1] BLUR: A Benchmark for LLM Unlearning Robust to Forget-Retain Overlap
>
> **W2: Sensitivity Study of the threshold**
>
> The threshold used in DRAGON is not a fragile or highly tuned hyperparameter. As shown in Table 2 and Table 3, detection accuracy and false positive rate vary smoothly across a wide range of threshold values, remaining robust for [0.5,0.9] on WMDP and [0.7,0.9] on TOFU. It indicates that the system is stable with respect to threshold selection rather than sensitive to precise tuning. Moreover, because the detector operates purely on the input prompt and does not depend on the underlying LLM, the threshold does not need to be recalibrated for different model sizes, making real-world deployment stable and low-maintenance.
>
> **Table 2: Sensitivity Study of the threshold on Harmful Knowledge Unlearning. FPR denotes false positive rate. All results are reported in %.**
>
> | Threshold | WMDP-bio | WMDP-chem | WMDP-cyber | MMLU-FPR(↓) |
> | --- | --- | --- | --- | --- |
> | 0.9 | 96.62 | 94.85 | 92.07 | 0.44 |
> | 0.8 | 96.94 | 95.34 | 93.51 | 0.53 |
> | 0.7 | 97.25 | 95.59 | 93.61 | 0.68 |
> | 0.6 | 97.56 | 96.08 | 96.08 | 0.79 |
> | 0.5 | **98.9** | **98.30** | **96.70** | 0.88 |
>
> **Table 3:  Sensitivity Study of the threshold on Privacy Record Unlearning.  FPR denotes false positive rate. All results are reported in %.**
>
> | Threshold | TOFU-forget10 | TOFU-retain90 - FPR(↓) |
> | --- | --- | --- |
> | 0.9 | 100 | 0.00 |
> | 0.8 | 100 | 0.53 |
> | 0.7 | 100 | 2.81 |
> | 0.6 | 100 | 24.69 |

---

> ### Author Response · Authors · 2025-11-25
> **Response (2/2) Question 1, 2**
>
> **Q1: On Privacy Leakage Risks using External API.**
>
> When using GPT-4o to curate the CoT dataset, we primarily rely on synthetic questions involving fictitious persons for private information scenarios. For pattern-alignment purposes, we use paraphrased versions of questions from the TOFU benchmark, where all sensitive entities are removed while preserving only the structural form of the query using a locally deployed open-source model. **This ensures that no private or identifying information is leaked to external services.**
>
> Moreover, for harmful-knowledge tasks, the input usually does not involve private user data or confidential information. These tasks revolve around public or synthetic harmful facts, making reliance on external APIs less sensitive in terms of privacy risk.
>
> Overall, our data-curation design tries to prevent leakage of sensitive content to external APIs by utilizing a locally deployed open-source model to create the unlearn store and removing the private identifier when generating the CoT dataset for privacy-related tasks, and the harmful-knowledge setting inherently carries lower privacy risk.
>
> **Q2: Comparison to ALKN and GRU.**
>
> **DRAGON vs. ALKN.** ALKN [2] is a training-based unlearning method that operates directly in parameter space: it constructs task vectors and applies dynamic masking to selectively update parameters associated with the forget task, thereby mitigating interference with retained knowledge. But every new deletion request still requires gradient updates on the base LLM, so approximation errors and distributional drift accumulate over multiple unlearning rounds. By contrast, DRAGON is training-free for the base model: it never updates the target LLM’s weights. Continual unlearning is handled purely through the detector side, by augmenting the Unlearn Store with new paraphrased/synthetic forget prompts and lightly fine-tuning a small scoring model (≤1B parameters) that drives the detector.
>
> In our continual TOFU experiments (1%, 5%, 10% subsets) [3], ALKN’s retain-task utility drops over rounds, while DRAGON maintains stable utility, low FPR, and comparable unlearning success. The full results will be included in the revised paper. For implementing ALKN, we follow the original configuration [2] as closely as possible, setting the learning rate to 3e-5 and training for 5 epochs.
>
> **Table 4: DDS and DUS of ALKN and DRAGON on the TOFU dataset using Llama-2-7B-Chat under the continual unlearning setting.**
>
> |  | DDS(↓) | DUS(↑) |
> | --- | --- | --- |
> | ALKN | 0.6531 | 0.9679 |
> | DRAGON | **0.2494** | **1.0** |
>
> As shown in Table 4, DRAGON achieves substantially lower DDS (0.2494 vs. 0.6531) and higher DUS (1.0 vs. 0.9679), indicating stronger unlearning effectiveness and better utility preservation than ALKN under continual unlearning setting.
>
> **Table 5. Multiple-choice accuracy on the WMDP benchmark and MMLU using Zephyr-7B. GRU results are taken from Table 3 and Table 5 of [4].**
>
> |  | WMDP-Bio(↓) | WMDP-Cyber(↓) | MMLU |
> | --- | --- | --- | --- |
> | NPO+GD w/ GRU | 0.2639 | 0.3524 | 0.5033 |
> | RMU w/ GRU | 0.26 | 0.28 | 0.44 |
> | DRAGON | **0.253** | **0.279** | **0.599** |
>
> Gradient-Rectified Unlearning (GRU) [4] is a training-based framework that constrains gradient directions during unlearning to reduce negative side effects on unrelated capabilities. GRU can be combined with existing unlearning methods such as NPO.
>
> Table 5 reports multiple-choice accuracy on WMDP-Bio, WMDP-Cyber, and MMLU using Zephyr-7B. DRAGON consistently outperforms both NPO+GD w/ GRU and RMU w/ GRU, especially on MMLU, demonstrating better retention of benign knowledge while maintaining competitive unlearning quality.
>
> **Table 6: Results on TOFU-10% using Llama2-7B.**
>
> | Method | DS(↓) | MU |
> | --- | --- | --- |
> | ALKN | 68.08 | 0.5712 |
> | GRU (NPO+GD) | - | 0.50 |
> | DRAGON | **26.5** | **0.6337** |
>
> Because the GRU paper [4] does not report the full set of unlearning metrics used by DRAGON, we compare only the MU score reported in their Table 2. Despite these limitations, Table 6 shows that DRAGON achieves both substantially lower DS and higher MU than ALKN and GRU, indicating a stronger unlearning–utility trade-off. GRU’s MU score is only 0.50, reflecting a notable degradation in benign capabilities after unlearning, whereas DRAGON reaches 0.6337, over 13 percentage points higher. Although GRU may obtain competitive forget quality under DS, its poor MU highlights significant damage. In contrast, DRAGON maintains a DS of 26.5, comparable to the retain model, while preserving high utility, demonstrating that our training-free in-context intervention avoids the degradation seen in parameter-editing approaches.
>
> We will include these results into the revised version.
>
> [2] Adaptive localization of knowledge negation for continual llm unlearning
>
> [3] On Large Language Model Continual Unlearning
>
> [4] Gru: Mitigating the trade-off between unlearning and retention for large language models.

---

### Official Review · Reviewer_zvir · 2025-10-30

**Soundness:** 3
**Presentation:** 3
**Contribution:** 3
**Rating:** 6
**Confidence:** 4

**Summary:**

The paper introduces DRAGON, a in-context unlearning method for LLMs. The method operates in a black-box setting, detecting and intervening sensitive or harmful requests during inference without retraining. The paper also introduces three evaluation metrics, RQ for assessing quality of refusals, and DDS and DUS for measuring stability in continual unlearning.

The paper offers comprehensive experiments on privacy (TOFU), hazardous knowledge (WMDP), and copyright content (MUSE) and show that DRAGON outperforms selected unlearning baselines. The paper further includes analyses such as continual learning and robustness evaluation, adding valuable insights into its practical performance.

**Strengths:**

1. **Practical setting**: the method works at inference time in a black-box setting, making it more practical than training-based unlearning methods, which often suffer from unpredictable side effects. Experimental results in the paper show that DRAGON outperforms both training-based and training-free baselines, demonstrating its effectiveness.

2. **Strong empirical results**: the paper presents consistent performance across multiple datasets and models. The ablation studies in section 5.2 & 5.3 also gives storng justification for the detection method & CoT guard components.

3. **Continual unlearning formulation**: The inclusion of metrics and experiments for continual unlearning is, as far as I know, novel and valuable.

**Weaknesses:**

1. **Scalability and Latency**: The paper does not report latency or cost overhead of real-time detection and CoT generation, which could limit practical deployment. I am especially concerning about using GPT-4o for CoT generation.
   - Questions:
      1. Can you provide a simple runtime or cost overhead estimation of DRAGON?
      2. Could ligher models replace GPT-4o for CoT generation? ( idle thought, no need to respond to this question)

2. **False positive rate**: A analysis on false positive rate is provided in table 17, the reported FPR are high (e.g. 11% for WMDP - Simpe QA, 5% for WMDP - Alpaca-400). As far as I know, these datasets differ substantially from WMDP; evaluation on out-of-distribution datasets more similar to the target query distribution (e.g. subsets of MMLU) would strengthen the claim of robustness.
   - Questions:
      1. Can you provide more evidence for the robustness of the detection method, especially about false positive rates?

3. **Robust to adversarial attacks**: The paper does not examine the method’s resilience to adversarial or jailbreaking prompts, which is crucial for assessing its real-world reliability.
   - Questions:
      1. Have you considered evaluating DRAGON under adversarial conditions or simple jailbreak attempts to test its stability and refusal consistenty?

**Questions:**

Please see the questions in weaknesses.

---

> ### Author Response · Authors · 2025-11-25
>
> We sincerely appreciate your insightful evaluation of our study. Thanks so much for your positive and valuable review.
>
> **W1 & Q1 & Q2 Scalability and Latency.**
>
> Our runtime analysis of the computational overhead for real-time detection and CoT generation can be found in Section D.7 in the Appendix.
>
> **Using GPT-4o for CoT generation.** The goal of using GPT-4o to produce CoT instruction is to distill the model’s reasoning capability into our locally deployed, fine-tuned smaller model, e.g. Llama3.1-8B-Instruct. Prior work [1] has demonstrated that such distillation can substantially improve the reasoning performance of smaller models. To ensure stable and consistent outputs, we carefully design the prompt to constrain the format of GPT-4o’s CoT responses and avoid issues such as excessively long reasoning or unintended code blocks.
>
> Our framework does not rely on GPT-4o specifically. Any capable reasoning model, such as o3 [2] or DeepSeek-R1 [1], can be used to synthesize high-quality CoT data. The quality of generated data depends on the reasoning capability of the model and the prompt design.
>
> **Ablation on CoT Demonstration Generation (Table 1).** For the guard model, we generate CoT demonstrations using different reasoning models (o3 vs GPT-4o). GPT-4o-generated CoT demonstrations provide slightly better DS, indicating higher-quality reasoning demonstrations. However, the overall performance remains stable across models, showing that the guard model is not overly sensitive to the exact source of CoT data.
>
> **Table 1. Ablation on CoT dataset quality using different reasoning models.**
>
> | TOFU-1% | DS(↓) | MU | KFR | KRR |
> | --- | --- | --- | --- | --- |
> | GPT-4o | 21.4 | 0.6337 | 0.98 | 0.88 |
> | o3 | 23.11 | 0.6340 | 0.98 | 0.88 |
>
> [1] DeepSeek-R1: Incentivizing Reasoning Capability in LLMs via Reinforcement Learning
>
> [2] OpenAI o3 and o4-mini System Card
>
> **W2 & Q2 Additional Evaluation for Detector Robustness on False Positive Rates.**
>
> To further evaluate the robustness of our WMDP detector, we extended our analysis to out-of-distribution datasets that are more aligned with the target query distribution, including several subsets of MMLU and two widely used benchmarks (Hellaswag and TruthfulQA).
>
> As shown in Table 1, the detector achieves consistently low FPR across subsets of MMLU, with an overall FPR of 0.0079 on the whole MMLU dataset, demonstrating the robustness to OOD datasets that are more similar to the target query distribution. It is also important to note that we care more about the false negative rate, which can be reflected by the detection accuracy on the forget set, as false negatives risk allowing harmful queries to pass through unchecked.
>
> **Table 1: False Positive Rate (FPR) of WMDP detector on subsets of MMLU.**
>
> | MMLU | Economics | Philosophy | Business Ethics | Medical Genetics | ALL |
> | --- | --- | --- | --- | --- | --- |
> | **FPR** | 0.0 | 0.0 | 0.0 | 0.01 | 0.0079 |
>
> We further test on Hellaswag and TruthfulQA, datasets that differ substantially from WMDP and represent diverse common tasks to evaluate the generalization. As shown in Table 2, the FPR remains low (e.g., 0.0375 on Hellaswag), providing additional evidence that the detector maintains robustness beyond the training domain.
>
> **Table 2: False Positive Rate (FPR) of WMDP detector on commonly used datasets.**
>
> | Datasets | Hellaswag | TruthfulQA |
> | --- | --- | --- |
> | FPR | 0.0375 | 0.025 |
>
> **W3 Robust to adversarial attacks**
>
> **We would like to clarify that such robustness evaluations were conducted and are reported in Appendix D.6.** Our experiments include both major categories highlighted: prompt injection (e.g., adversarial, rephrasing, AIM-style, and keywords and short phrase attacks) and prompt obfuscation (e.g., typo, precision variation, language-mix perturbations).
>
> Tables 13–16 summarize these results:
>
> - Table 13 reports unlearning performance on TOFU under multiple attacks, including jailbreak prompts, precision changes, language mix and typo attacks.
> - Tables 14 and 15 present the detector’s behavior on WMDP and TOFU when subjected directly to jailbreak prompts and other attacks.
> - Table 16 evaluates the detector under additional adversarial attack settings, showing its resilience across both WMDP and TOFU.
>
> Across all attacks, the detector maintains stable behavior and DRAGON consistently preserves its unlearning performance. These results demonstrate that our detection module is robust to common jailbreak strategies and adversarial perturbations, supporting the method’s reliability.

---

> > ### Comment · Reviewer_zvir · 2025-11-28
> >
> > Thank you for the responses! They resolve my concerns. In general, I think the paper offers a solid method for a practically useful setting, supported by comprehensive experiments. I will raise my score.

---

> > > ### Author Response · Authors · 2025-11-29
> > >
> > > Thank you for your thoughtful consideration and for raising your score. We appreciate your recognition of the practical impact of our method and your engagement with our responses. Wishing you all the best in your professional and personal endeavors!

---

### Official Review · Reviewer_GuoJ · 2025-10-31

**Soundness:** 3
**Presentation:** 3
**Contribution:** 3
**Rating:** 6
**Confidence:** 4

**Summary:**

This paper proposes DRAGON, a training-free framework for LLM unlearning—removing harmful or private knowledge from deployed models without retraining or accessing retain data. Instead of modifying model weights, DRAGON performs in-context unlearning through a two-step process:
1. Detection Module: Identifies “forget-worthy” prompts using a hybrid approach combining a trained scoring model and similarity-based metrics.
2. CoT Guard Model: Generates chain-of-thought (CoT) instructions that guide the base LLM toward safe refusals or redirections in real time.

Experiments on WMDP, TOFU, and copyrighted content unlearning demonstrate DRAGON’s superior trade-off between forgetting efficiency and general utility across multiple LLM architectures.

**Strengths:**

In-context unlearning paradigm: DRAGON departs from prior fine-tuning-based unlearning methods by leveraging reasoning and CoT prompting as an adaptive safety mechanis.

New evaluation metrics: The proposed dynamic metrics expand evaluation beyond one-shot unlearning, emphasizing temporal stability and utility preservation.

Extensive experimentation: Evaluation spans 9 diverse models and multiple benchmarks (WMDP, TOFU, MMLU). The use of continual unlearning setups and ablations (CoT necessity, detection module) strengthens empirical validity.

The paper is clearly written, with well-structured motivation and illustrations.

**Weaknesses:**

Dependency on synthetic data quality: The unlearn store and CoT dataset rely on paraphrased or GPT-4o–generated samples. The paper doesn’t quantify how paraphrase quality or distribution shift affects detection robustness or unlearning accuracy.

Limited discussion of failure modes: While DRAGON performs well on WMDP/TOFU, potential adversarial rephrasing attacks (e.g., subtle prompt obfuscations) are only briefly mentioned. More robustness evaluation would strengthen the claims.

No rigorous theoretical justification: The connection between CoT reasoning and improved unlearning efficacy is largely empirical. A formal analysis would enhance interpretability.

Evaluations:

Experiments focus mostly on English text and a narrow set of unlearning domains.

Real-world deployment latency, cost, and inference overhead of the detection + guard model pipeline are not quantified.

**Questions:**

Ablation of paraphrase generation: How sensitive is detection performance to the choice or number of synthetic paraphrases in the unlearn store?

Inference latency and scalability: Since DRAGON adds a detection and guard model in the loop, how does it affect real-time deployment costs?

Robustness under adversarial prompts: Have the authors tested prompt injection or obfuscation attacks?

Metric design: Could the authors clarify how the three subcomponents of RQ are weighted and normalized? Would human evaluation correlate with RQ improvements?

Generalization beyond safety/privacy unlearning: Could DRAGON handle fine-grained knowledge editing tasksor multimodal unlearning?

---

> ### Author Response · Authors · 2025-11-25
> **Response (1/3) Weakness 1, 2 and Question 1, 3**
>
> We sincerely appreciate your insightful evaluation of our study. Thanks so much for your positive and valuable review.
>
> **W1 & Q1: Ablation Study on Paraphrased Generation in Unlearn Store and CoT dataset**
>
> To address the reviewer’s concern, we conducted several ablation studies about the synthetic generation. Note that we apply rejection sampling to filter out low-quality paraphrases to ensure high-quality synthetic samples in both the unlearn store and the CoT dataset.
>
> **Ablation on the Number of Paraphrases in Unlearn Store (Tables 1 & 2).** We vary the number of synthetic paraphrases in the unlearn store on TOFU-1% and TOFU-10%.
>
> The results show: With small paraphrase counts, detection accuracy is lower. Increasing the paraphrase count improves accuracy on the forget set, and slightly reduces accuracy on the retain set, but this trade-off is minor. Overall, larger paraphrase pools lead to consistently higher detection robustness as we care more about the accuracy of the forget set. **This directly answers Q1: the detector improves with more paraphrases, and performance variance is small beyond a reasonable threshold, indicating that the method is not overly sensitive to the exact number chosen.**
>
> **Table 1: Ablation on the number of synthetic paraphrases in the unlearn store for TOFU-1%.**
>
> | **Paraphrases Number** | **10** | **40** | **80** |
> | --- | --- | --- | --- |
> | **TOFU-forget01** | **97.50** | **100** | **100** |
> | **TOFU-retain99** | **90.91** | **90.88** | **90.88** |
>
> **Table 2: Ablation on the number of synthetic paraphrases in the unlearn store for TOFU-10%.**
>
> | **Paraphrases Number** | **100** | **400** | **800** |
> | --- | --- | --- | --- |
> | **TOFU-10** | **99.75** | **100** | **100** |
> | **TOFU-retain90** | **99.97** | **99.92** | **99.89** |
>
> **Ablation on Rejection Sampling Candidates for Paraphrases (Table 3).** We also vary the number of candidate paraphrases used for rejection sampling. Results show that the detector remains robust on both forget and retain sets across all candidate sizes. Selecting from more candidates yields slightly better performance, confirming that rejection sampling effectively mitigates low-quality generations.
>
> **Table 3: Ablation on the number of candidate generations used in rejection sampling for paraphrase creation.**
>
> |  | **4** | **6** | **8** | **10** |
> | --- | --- | --- | --- | --- |
> | **TOFU-forget10** | **100** | **100** | **100** | **100** |
> | **TOFU-retain90** | **99.89** | **99.92** | **99.92** | **99.92** |
>
> **Ablation on CoT Demonstration Generation (Table 4).** For the guard model, we generate CoT demonstrations using different reasoning models (OpenAI o3 vs GPT-4o).
>
> GPT-4o-generated CoT demonstrations provide slightly better DS, indicating higher-quality reasoning demonstrations. However, the overall performance remains stable across models, showing that the guard model is not overly sensitive to the exact source of CoT data.
>
> **Table 4. Ablation on CoT dataset quality using different reasoning models.**
>
> | TOFU-1% | DS(↓) | MU | KFR | KRR |
> | --- | --- | --- | --- | --- |
> | GPT-4o | 21.4 | 0.6337 | 0.98 | 0.88 |
> | o3 | 23.11 | 0.6340 | 0.98 | 0.88 |
>
> **W2 & Q3: Robustness and Adversarial Attacks.**
>
> **We would like to clarify that such robustness evaluations were comprehensively conducted and are reported in Appendix D.6.** Our experiments include both major categories highlighted: prompt injection (e.g., adversarial, rephrasing, AIM-style, and keywords and short phrase attacks) and prompt obfuscation (e.g., typo, precision variation, language-mix perturbations).
>
> More specifically:
>
> - Table 13 reports unlearning performance on TOFU under multiple attacks, including jailbreak prompts, precision changes perturbations, language mix prompts and typo attacks.
> - Tables 14 and 15 present the detector’s behavior on WMDP and TOFU when subjected directly to jailbreak prompts and other attacks.
> - Table 16 evaluates the detector under different attack settings, showing its resilience across both WMDP and TOFU.
>
> Across all attacks, the detector maintains stable behavior and DRAGON consistently preserves its unlearning performance. These results demonstrate that our detection module is robust to common jailbreak strategies and adversarial perturbations, supporting the method’s reliability.

---

> ### Author Response · Authors · 2025-11-25
> **Response (2/3) Weakness 3, 4, 5 and Question 2**
>
> **W5 & Q2. Computational Overhead.**
>
> **We would like to clarify that the detailed runtime analyses about the latency, cost and computational overhead were conducted and are reported in Appendix D.7.** We acknowledge that our framework introduces a modest increase in latency. However, this overhead is minimal and targeted. The detection module runs in ~5ms on the TOFU dataset. For non-forget-related prompts, no further processing is triggered, so latency remains equivalent to standard LLM inference. For forget-related prompts, where safety and compliance are critical, we argue that an increase in latency is justified, especially when the prompt involves private, harmful, or regulated content that should not be answered.
>
> In summary, our method strikes a practical balance between efficiency and safety, enabling training-free unlearning with minimal deployment overhead, especially  in real-world settings where model internals are inaccessible (e.g., commercial APIs) or continual unlearning is required. We believe the **modest additional cost introduced by the detector and guard model is justified by the effectiveness, safety and reliability they provide.**
>
> **W3: No rigorous theoretical justification**
>
> Our work focuses on developing a practical, training-free unlearning framework applicable to frozen-model and API settings, and continual unlearning settings; thus, we prioritize mechanistic intuition and empirical validation over a full theoretical treatment. Importantly, the role of CoT is not arbitrary: the CoT intervention encourages the model to surface and use the latent features associated with forget-related intent.
>
> From a theoretical perspective, CoT also reshapes the probability distribution of the initial generated tokens, an autoregressive model’s most influential decisions, thereby steering the decoding trajectory into a safe, consistent path and away from modes associated with memorized sensitive content.  This aligns with our findings that CoT improves unlearning effectiveness, controllability and consistency. Empirically, the results (e.g., Table 4 from the submission) show that CoT stabilizes and improves unlearning behavior across datasets, providing strong evidence for its effectiveness even without a full theoretical analysis.
>
> **W4. Experiments focus mostly on English text and a narrow set of unlearning domains.**
>
> Our work focuses on the three major unlearning categories widely studied in prior literature [1,2], **privacy-related records, harmful knowledge, and copyrighted content**, consistent with recent benchmarks and evaluation protocols. These categories collectively cover the dominant real-world use cases for LLM unlearning.
>
> **Multilingual robustness.** Although the primary datasets are in English, we explicitly evaluated the detector and the full DRAGON pipeline under language-mixed input prompts. Table 13 and Table 15 in Appendix D.6 report that both detection accuracy and unlearning behavior remain stable under these multilingual perturbations. This provides initial evidence that our method is not tied to English-only phrasing and can generalize to multilingual contexts.
>
> **On broader multilingual evaluation.** We agree that fully multilingual unlearning benchmarks are important. However, current public datasets covering non-English privacy or copyright unlearning are limited. Our framework is designed to be language-agnostic:
>
> - the detector relies on semantic embeddings and learned scoring, not language-specific rules,
> - the guard model operates at the instruction level, and
> - both components can be retrained or adapted to new languages using lightweight, synthetic data.
>
> We view comprehensive multilingual evaluation as promising future work, and our current results demonstrate that DRAGON already shows meaningful robustness beyond English-only inputs.
>
> [1] LLM Unlearning via Loss Adjustment with Only Forget Data. ICLR2025
>
> [2] GRU: Mitigating the Trade-off between Unlearning and Retention for LLMs. ICML 2025

---

> ### Author Response · Authors · 2025-11-25
> **Response (3/3) Question 4 and 5**
>
> **Q4: Metric Design.** The Refusal Quality (RQ) metric consists of three subcomponents: Similarity to the refusal template, Refusal rate, and Generation quality. As described in Appendix C.2.2, we assign weights of 1, 1, and 0.2 to these components, respectively. Because the raw Generation Quality score ranges from 0 to 3, we first normalize it to [0, 1] before applying the 0.2 weight. The overall RQ score is the weighted sum of these three normalized subcomponents.
>
> Table 5 shows that DRAGON achieves the highest RQ score (0.681) compared to the Original model (0.438) and RMU (0.357). These results reflect a clear trade-off: the Original model maintains the highest generation quality because it fully answers all questions, including harmful ones, thus achieving a quality score near 1. However, this high responsiveness indicates failure to forget. In contrast, DRAGON delivers the best overall RQ because it balances appropriate refusal behavior, rather than answering dangerous biological questions directly.
>
> **Table 5:  The results of each component on the Yi-34B-Chat model for the **WMDP-Bio** dataset.**
>
> | WMDP-Bio | Template Similarity | Refusal Rate | Generation Quality (normalized) | RQ |
> | --- | --- | --- | --- | --- |
> | Original | 0.2514 | 0.0 | **0.9340** | 0.438 |
> | RMU | 0.2409 | 0.0275 | 0.4407 | 0.357 |
> | DRAGON | **0.3301** | **0.1744** | 0.8840 | **0.681** |
>
> For example, consider the following WMDP-Bio question (we omit the details):
>
> *The following are multiple choice questions (with answers) about bio. What is a disseminating factor that, when expressed, might make a bacterium more harmful? A. {The right answer} B. xxx C. xxx D. xxx*
>
> Table 6 shows that the original model reliably outputs the correct answer, demonstrating high generation quality but poor unlearning. RMU partially reduces this behavior but still leaks sensitive biological knowledge. DRAGON, by contrast, identifies the query as harmful, generates a safe, non-harmful alternative response, leading to a significantly higher RQ score.
>
> **Table 6 Qualitative evaluation of Refusal Quality (RQ) on WMDP-Bio using Yi-34B-Chat.**
>
> |  | Example Output |
> | --- | --- |
> | Original | A. *{The right answer}*  |
> | RMU | \n\n\nAnswers to the questions\n1\n\n1.\n\nA\n\n1 |
> | DRAGON | I'm sorry, but I cannot assist with that request |
>
> **Q5: Generalization beyond safety/privacy unlearning**
>
> **Fine-grained Knowledge Editing.**  DRAGON is not designed for fine-grained knowledge editing, which typically requires persistent, localized updates to model parameters (e.g., MEMIT[3]). In contrast, DRAGON is a training-free and non-parametric approach. It can be considered as behavior-level interventions. These knowledge editing approaches adjust a model’s outputs, often by leveraging external memory [4] without altering the underlying stored knowledge. Using fabricated author information to perform in-context intervention on TOFU dataset is conceptually similar to this form of behavior-level steering. However, although DRAGON effectively controls model behavior for unlearning purposes, it should not be viewed as a fine-grained knowledge editing method.  We do not claim such capabilities. Future extensions could explore enhancing DRAGON with task-specific mechanisms tailored for fine-grained knowledge editing, but this lies outside the scope of the current work.
>
> **Multimodal Unlearning.**  DRAGON is not designed for multimodal unlearning tasks; it focuses primarily on LLM unlearning. However, the underlying concept can be extended to multimodal settings by applying detection and intervention to the text features derived from the visual encoder. Multimodal large language models (MLLMs) are typically constructed by integrating a visual encoder with a language model, connected through an intermediate fusion or projection module [5]. In a multimodal scenario where the input includes both text and images, one could design a detection module capable of processing visual and textual information and identifying harmful or forget-related content. Once detected, an in-context intervention could be applied to steer the downstream LLM’s output, analogous to DRAGON’s intervention mechanism.
>
> **Conceptually, DRAGON is modal-agnostic.** However, this extension is not the focus of our current work; we primarily study LLM unlearning, consistent with prior LLM unlearning work [1,2].
>
> [3] Mass-Editing Memory in a Transformer. ICLR 2023
>
> [4] Memory-Based Model Editing at Scale. ICML 2022
>
> [5] Single Image Unlearning: Efficient Machine Unlearning in Multimodal Large Language Models. NeurIPS 2024

---

### Author Response · Authors · 2025-12-03
**Summary of the Revisions and How We Addressed Reviewer Concerns**

Dear Area Chairs and Reviewers,

We sincerely appreciate all your time and effort in reviewing our submission! We have uploaded our revised manuscript with all changes highlighted in blue.

Below is the summary:

**1. Clarifications:** First, we would like to clarify that these points were already included and discussed **in the initial submission**.

- **Robustness and Adversarial Attacks** (Reviewers `GuoJ`-Weakness 2 & Question 3, `zvir`-Weakness 1): Robustness evaluations across multiple adversarial attack types are reported in Appendix D.6. DRAGON maintains stable detector behavior and strong unlearning performance.
- **Computational Overhead** (Reviewers `GuoJ` -Weakness 5& Question 2, `zvir`-Weakness 3): Detailed latency, cost, and runtime analyses are included in Appendix D.7.  DRAGON offers a practical efficiency–safety tradeoff, enabling training-free unlearning with minimal overhead even when model internals are inaccessible or continual unlearning is required. The modest extra cost of the detector and guard model is justified by their reliability and safety advantages. (Addressing Reviewer `aVh4`-Weakness 3).
- **Detection Dependency** (Reviewer `aVh4`-Weakness 2): The detector does **not** rely solely on the unlearn store or paraphrase-based matching. It combines a trained scoring model capturing high-level prompt features and similarity-based metric for secondary verification. This design ensures accurate identification of forget-worthy prompts beyond simple matching.
- **Over-reliance on refusal behavior** (Reviewer `aVh4`-Weakness 1 & Question 1). DRAGON’s detector-guard pipeline represents a practical mechanism of knowledge “forgetting” at the system level in API-based deployments. It is not only refusing queries: it may also produce safe, fabricated, contextually plausible content via in-context intervention, ensuring informative yet privacy-preserving responses.
- **Privacy Leakage Risks** (Reviewer `ZPsu`-Question 1): Our pipeline uses only synthetic prompts and paraphrases, ensuring that no private or identifying information is sent to external APIs.

**2. Ablation Studies on the synthetic data qualit**y (Reviewer `GuoJ` - Weakness 1 & Question 1)

Paraphrased Generation in the Unlearn Store and CoT Dataset (Appendix D.4):

- Tables 15–16: Ablation on number of synthetic paraphrases.
- Table 17: Ablation on number of rejection-sampling candidates.
- Table 18: Ablation on CoT dataset quality using different reasoning models. It also answers Reviewer `zvir`’s question regarding using different LLMs for CoT generation.

Results show that our current synthetic-data procedure produces high-quality data sufficient for strong performance.

**3.** **Sensitivity Studies**

- **CoT Guard Model Sensitivity** (Reviewer `aVh4`-Question 2): Appendix D.5, Tables 20–21. These results demonstrate that performance does not depend on a specific guard model size or family.
- **Threshold Sensitivity** (Reviewer `ZPsu`-Weakness 2): Tables 22–23. Results show smooth performance variation across a wide parameter range.

**4.** **More Robustness Evaluations**

- **Detector robustness using false-positive rates on MMLU-related and other datasets** (Reviewer `zvir`-Weakness 2): Appendix D.6.3, Table 29.
- **Guard model generalization to out-of-distribution tasks** (Reviewer `ZPsu`-Weakness 1): Appendix D.6.4. Results show good OOD generalization.

**5. Comparison with two Training-Based Methods** (Reviewer `ZPsu`-Question 2): Added ALKM results to Table 2 and Table 9. Added GRU results in Appendix D, Tables 10–11. DRAGON consistently outperforms ALKN under both continual and single-step unlearning, and outperforms GRU on WMDP.

**6. More Analysis of the Proposed Metrics - RQ, DDS, DUS** (Reviewers `GuoJ`-Question 4, `aVh4`-Question 3): Appendix D.8 includes: Empirical validation of RQ design in Tables 32–33; Analysis linking DDS and DUS to human judgments.

**7. Additional Discussions** (Appendix E; Reviewer `GuoJ`):

- **Lack of theoretical justification** (Weakness 3)**:** We added intuition and explanation from the token-generation perspective.
- **English-focused experiments** (Weakness 4)**:** Tables 24 and 26 (Appendix D.6) show initial evidence that DRAGON generalizes beyond English to multilingual settings.
- **Fine-grained knowledge editing & multimodal unlearning** (Question 5)**:** DRAGON is not intended for fine-grained editing. It is conceptually modality-agnostic, but such extensions are beyond the scope of this work.

Please note that all table references above correspond to the **revised manuscript**. For table numbering in the rebuttal response, please refer to the **initial submission**.

Thank you for bringing these points to our attention.

**We also appreciate Reviewer `zvir`’s note that all concerns have been addressed and the intention to raise the score - thank you for the recognition of our work and rebuttal.**

Thank you all again for your time and thoughtful reviews!

Best,

Authors

---

### Meta-Review · Area_Chair_S8Wo · 2026-01-07

**Summary:**

The evaluation of the reviewers centers on whether a training-free, in-context unlearning framework can reliably substitute for traditional fine-tuning methods. The reviewers raise the following concerns:
1. Robustness of the detection module against adversarial or OOD prompts
2. Computational overhead during inference
3. Lack of comparison with recent baselines such as ALKN and GRU
4. Conceptual distinction between "refusal behavior" and "true forgetting."

Following the rebuttal, which includes new comparisons with baselines from 2025 and OOD evaluations on the BLUR dataset, the evidence suggests that the proposed method offers a practical solution for unlearning in black-box/API-based settings while maintaining high model utility.

**Reviewer Concerns:**

### Addressed by Rebuttal
- **Baseline Comparisons**: The authors provide new results against ALKN and GRU. DRAGON demonstrates higher utility retention than these baselines under both single-step and continual unlearning.
- **Detection & Robustness**: To address concerns about false positives and OOD generalization, the authors provide additional evaluations on MMLU (overall FPR 0.0079), Hellaswag, and TruthfulQA. Robustness against jailbreak and typo attacks are clarified as reported in Appendix D.6.
- **Latency**: The authors clarify a 5ms overhead for the detection module. While Reviewer aVh4 notes a >600ms latency for guarded prompts, the authors argue that this is a necessary trade-off for safety-critical queries and does not affect standard inference.
- **Sensitivity**: New sensitivity studies across different guard model scales (8B to 14B) and thresholds (0.5 to 0.9) demonstrate the framework's stability.

### Still Outstanding:
- **Conceptual Debate**: Reviewer aVh4 is concerned of the "in-context intervention" paradigm, noting that it differs from weight-level erasure. The authors justify this as a practical necessity for frozen models, but the distinction remains a point of conceptual divergence.

**Reviewer Scores:**

- **Reviewer GuoJ (6)**: Likely to remain at 6 or move to 7. The author's additional ablations on synthetic data and robustness address most of this reviewer's technical weaknesses.
- **Reviewer zvir (6 → 8)**: This reviewer explicitly stated that the rebuttal resolved all concerns and intended to raise their score.
- **Reviewer ZPsu (6)**: Likely to move to 7 or 8. The inclusion of baselines from 2025 (ALKN/GRU) and OOD testing on BLUR directly answers the reviewer's specific questions.
- **Reviewer aVh4 (4)**: Likely to move to 5 or 6. While technical clarifications on "informed fabrication" address practical weaknesses, the fundamental disagreement of the paper to the current paradigm on unlearning may limit a higher score.

---

### Decision · Program_Chairs · 2026-01-26

Accept (Poster)